# Total Flavonoid Contents and the Expression of Flavonoid Biosynthetic Genes in Breadfruit (*Artocarpus altilis*) Scions Growing on Lakoocha (*Artocarpus lakoocha*) Rootstocks

**DOI:** 10.3390/plants12183285

**Published:** 2023-09-16

**Authors:** Yuchan Zhou, Steven J. R. Underhill

**Affiliations:** Australian Centre for Pacific Islands Research, University of the Sunshine Coast, Sippy Downs, QLD 4556, Australia

**Keywords:** breadfruit (*Artocarpus altilis*), dwarfing, rootstock, flavonoids, chalcone synthase gene, bifunctional dihydroflavonol 4-reductase gene, lakoocha (*Artocarpus lakoocha*)

## Abstract

Breadfruit (*Artocarpus altilis*) is a traditional fruit tree of 15–30 m height in the tropics. The presence of size-controlling rootstock in the species is not known. A small tropical tree species, lakoocha (*Artocarpus lakoocha*), was recently identified as a potential vigor-controlling rootstock, conferring over a 65% reduction in breadfruit tree height. To better understand the intriguing scion/rootstock interactions involved in dwarfing, we investigate flavonoid accumulation and its regulation in breadfruit scions in response to different rootstocks. To this end, we isolated a chalcone synthase cDNA, *AaCHS*, and a full-length bifunctional dihydroflavonol 4-reductase cDNA, *AaDFR*, from breadfruit scion stems. The expression of both *AaCHS* and *AaDFR* genes was examined over the period of 16 to 24 months following grafting. During the development of the dwarf phenotype, breadfruit scion stems on lakoocha rootstocks display significant increases in total flavonoid content, and show upregulated *AaCHS* expression when compared with those on self-grafts and non-grafts. There is a strong, positive correlation between the transcript levels of *AaCHS* and total flavonoid content in scion stems. The transcript levels of *AaDFR* are not significantly different across scions on different rootstocks. This work provides insights into the significance of flavonoid biosynthesis in rootstock-induced breadfruit dwarfing.

## 1. Introduction

Fruit trees with reduced stature decrease production costs and allow for high-density planting. In many of these species, dwarfing rootstocks play an important role in vigor control. Breadfruit (*Artocarpus altilis* (Parkinson) Fosberg) is a traditional fruit tree of 15–30 m height found in Oceania. Primarily grown as an energy food, which is a source of complex carbohydrates, vitamins, and minerals, the species is regarded as a food-security crop in the tropics [1]. However, the presence of size-controlling rootstock in the species is not known. In the same genus of *Artocarpus*, a small tropical tree species, lakoocha (*Artocarpus lakoocha*), was recently identified as a potential vigor-control rootstock conferring over a 65% reduction in breadfruit tree height when compared with those on self-graft and own-rooted standard breadfruit trees [2]. Importantly, the assessment also indicated that the rootstock-induced dwarf phenotype in breadfruit trees on lakoocha rootstocks was not associated with graft incompatibility [2]. The intriguing scion/rootstock communications whereby lakoocha rootstock imparts size-control over breadfruit scions have not yet been investigated.

While rootstock-induced dwarfism has been extensively investigated in many species, its underlying mechanism is still not well understood. Several models have been suggested to elucidate rootstock-induced dwarfing. These include a decrease in water and solute transport over graft union, disruption to carbohydrate allocation and altered hormone signaling between scions and rootstocks [3,4]. The dwarfing effects of apple rootstock M.9 reduced auxin transport, and they increased export of abscisic acid (ABA) when compared with a vigorous rootstock (MM.106) [5]. It was also proposed that this reduction in basipetal auxin transport limited root growth, thereby restricting the amount of root-produced cytokinin supplied to scions [6,7]. Transcription profiles have revealed growth reduction in apple dwarfing interstocks, involve interaction of sugar metabolism and network of hormone signalling, including auxin, cytokinin, ABA, and gibberellin (GA) [8]. Interstock-induced dwarfism in sweet persimmon also involved lower levels of auxin and GA [9]. Transcriptome profiling also found breadfruit dwarfing through marang rootstocks was associated with downregulation of genes for signalling components of auxin and gibberellin [10]. It was shown that dwarfing rootstocks significantly increased the ratio of cytokinin/auxin levels, but decreased the GA content in scions of ‘Yunnan’ quince [11]. Similarly, the vigorous rootstock ‘Canton lemon’ had a higher level of auxin compared to those in dwarfing rootstock ‘Fragrant orange’ [12]. Furthermore, application of an auxin transport inhibitor, 1-N-naphthylphthalamic acid, to vigorous rootstocks reduced auxin transport, and conferred similar dwarfing effects on scions as those observed in drawing rootstocks, leading to significant reductions in shoot growth [13]. Due to the inhibitory role of flavonols in polar auxin transport [14], there is considerable evidence to suggest that flavonoids have a role in rootstock-induced dwarfing [10,15,16]. Dwarfing rootstocks were shown to cause over-accumulation of flavonoids in scions of apple [15] and pear [16]. Several flavonoid metabolites were significantly upregulated in mandarin scions grafted onto dwarfing rootstock ‘Flying Dragon’ when compared with those on vigorous rootstocks [17]. Concentrations of flavonoids procyanidin B2 and procyanidin B3 were remarkably higher in pear scions growing on dwarfing interstocks [16]. It was also discovered that dwarfism in sweet cherry through dwarfing rootstocks was caused by early termination of terminal shoot growth, which involved upregulation of a gene encoding a MYB protein involved in the early stages of flavonol biosynthesis [18]. Transcriptomic analysis also revealed upregulation of genes in the flavonoid biosynthesis pathway in scions on dwarfing-related rootstocks, such as M.9 [15] and marang [10]. Along with reduced polar auxin transport, flavonol-overproducing mutants often display dwarf phenotypes [14,19].

Flavonoids represent a class of polyphenolic metabolites with major roles in stress and defense signaling, cell growth, and differentiation [20,21]. Flavonoids are grouped into flavanols, flavandiols, flavanones, dihydroflavonols, flavones, isoflavones, and anthocyanidins, all bearing a common chemical skeleton of C6-C3-C6 consisting of two aromatic rings (A and B) and a heterocyclic ring (C) with one oxygen atom [22]. Biosynthesis of flavonoids in plants has been well characterized [20]. Condensation of *p*-coumaroyl-CoA and malonyl-CoA catalyzed by chalcone synthase (CHS) is the first committed step in the flavonoid biosynthesis pathway (Figure 1). This reaction produces a precursor for synthesis of a large number of flavonoid derivatives in the downstream pathway [22,23]. CHS is therefore a gatekeeper controlling carbon flux from the general phenylpropanoid pathway into the flavonoid pathway [24]. It has been shown that expression of *CHS* genes is significantly correlated with the accumulation of total flavonoids in many species [23,25]. Over-expression of *CHS* genes was reported to enhance flavonoid biosynthesis, while silencing of the *CHS* gene was associated with decreased flavonoid content [23,26,27]. Further into the flavonoid pathway (Figure 1), a committed step leading to the biosynthesis of anthocyanin and proanthocyanidin is catalyzed by the bifunctional dihydroflavonol 4-reductase (DFR) [22,28]. DFR is a key enzyme responsible for the NADPH-dependent reduction of the colorless dihydroflavonols into flavan-3,4-diols (leucoanthocyanidins) [29,30,31]. It has been reported that the expression levels of *DFR* genes are highly correlated with anthocyanin accumulation [32,33,34]. Moreover, the temporal and spatial expression of flavonoid biosynthetic genes, including *CHS* and *DFR* genes, is controlled by a plethora of transcription regulators, such as the MYB-bHLH-WDR protein complex, which are involved in responses to numerous developmental and environmental signals during growth and development [21,35].

Both *CHS* and *DFR* genes have been cloned and characterized from diverse species, such as *CHS* from mulberry (*Morus atropurpurea*) [36], grapevine [37], apple (Malus × domestica) [38], citrus (*Citrus clementina* Oroval × *Citrus sinensis* ‘Moro’) [23], and soybean [39], and *DFR* from orange (*Citrus sinensis*) [29], tea (*Camellia sinensis*) [40], *Populus trichocarpa* [33], *Rhododendron delavayi* [34], *Lotus japonicus* [28], *Medicago truncatula* [32], and *Freesia hybrid* [31]. CHS is generally encoded by gene family of various sizes [23], while single or multiple copies of *DFR* genes have been reported [28]. Partial sequences of a *CHS* gene and a *DFR* gene were previously identified in breadfruit species, with both genes expressed in scion stems in marang rootstocks [10]. It remains unknown whether they play any role in scion flavonoid biosynthesis or rootstock responses.

In the current study, two key flavonoid biosynthetic genes, *AaCHS* and *AaDFR*, are cloned in full-length in breadfruit scions and further investigated for their expression profiles in correlation with the total flavonoid contents in scions growing on different rootstocks. The involvement of flavonoid accumulation and the flavonoid biosynthetic genes in the development of rootstock-induced dwarfing in breadfruit trees is also discussed. This information improves our understanding of the molecular mechanisms underlying breadfruit dwarfing in interspecific rootstocks.

## 2. Results

### 2.1. Effects of Rootstocks on Height and Total Flavonoid Content in Breadfruit Scions

Breadfruit scions on lakoocha rootstocks exhibited shorter stature, with stem height reduced by 68.6% in the period up to 12 months following grafting, and by 67.7% in the period up to 24 months following grafting when compared with those on self-grafts in the same period (Figure 2a and Appendix A). There was no significant difference in scion height between self-grafts and non-grafts. These results were in agreement with previous findings for the period up to 21 months following grafting [2].

Total flavonoid contents were measured in scion stems every 2 months in the period from 16 to 24 months after grafting. Although there was no significant change in flavonoid levels across this period for any rootstock type (Figure 2), our results showed that scion stems on lakoocha rootstocks displayed significantly higher flavonoid contents at four out of five time points, with increases of 73.3%, 69.9%, 88.0%, and 60.5%, respectively, at 18, 20, 22, and 24 months, when compared with self-grafts at the same time points (Figure 2). Flavonoid levels between self-grafts and non-grafts showed no significant differences.

### 2.2. Isolation and Expression Analysis of the Chalcone Synthase Gene in Breadfruit Scions

Chalcone synthase (CHS) is considered the first committed enzyme in the pathway of flavonoid biosynthesis [24]. The partial sequence of a *CHS* gene, *AaCHS*, was found in breadfruit stems [10]. To confirm the presence of this gene, full-length *AaCHS* was isolated using 5′ and 3′ RACE (see Methods). Sequence analysis showed that the predicted protein sequence of *AaCHS* bore all the hallmarks of functional CHS families (Figure 3a). These included the presence of a strictly conserved catalytic site consisting of four residues: Cys164, His303, Asn336, and Phe215, corresponding to the crystal structure of *Medicago sativa* MsCHS2 [41], and a characteristic CHS signature, GFGPG, which modulates stereochemistry processes in the cyclization reaction [42]. Phylogenetic analysis (Figure 3b) showed the predicted AaCHS was closely related to the *Morus notabilis* homolog, MnCHS2 (Genbank acc. XP_024027307), with 72% identity and the *Morus alba* homolog, MaCHS (Genbank acc. AOV62760), with 69% identity.

Transcript levels of *AaCHS* were analyzed using quantitative real-time RT-PCR in scion stems grafted onto different rootstocks (Figure 4). Over the period from 16 to 24 months after grafting, it was shown scions on lakoocha rootstocks had significantly higher levels of *AaCHS* expression in stems at all the time points measured, with an overall increase ranging from 84.9% to 387.5% when compared with those on self-grafts. Expression levels between self-grafts and non-grafts showed no significant difference (Figure 4).

### 2.3. Isolation and Expression Analysis of the Dihydroflavonol 4-Reductase Gene in Breadfruit Scions

Dihydroflavonol-4-reductase (DFR) is a pivotal enzyme in the flavonoid pathway and leads to the biosynthesis of anthocyanin and condensed tannin. In the current study, a full-length DFR gene, *AaDFR*, was isolated using 5′ and 3′ RACE based on a partial sequence previously detected in breadfruit tissues [10]. Sequence analysis showed that the predicted protein for *AaDFR* had all the characteristic typical of functional DFRs. These included the presence of a highly conserved putative NADPH-binding motif in the N-terminus, as well as a presumed substrate-binding domain (Figure 5a). In addition, an aspartic acid at position 134 suggested AaDFR might have a substrate preference for dihydroquercetin (DHQ) and dihydromyricetin (DHM) over dihydrokaempferol (DHK) [43]. Phylogenetic analysis (Figure 5b) showed the predicted protein encoded by *AaDFR* was closely related to MaDFR (Genbank acc. AHB19302) from *Morus alba*, sharing 85% identity. As expected, the *AaDFR* gene was also clustered under a subclade of eudicots.

Scion stems growing on different rootstocks were examined for the expression of *AaDFR*. Over a 9-month period, fluctuations in the transcript levels of *AaDFR* were observed in all samples, but differences between plants growing on different rootstocks at any time were not significant (Figure 6).

## 3. Discussion

Breadfruit plants with lakoocha rootstocks were about one-third of the standard height by the end of 24 months following grafting (Figure 1 and Appendix A), in agreement with previous findings showing the potential of lakoocha rootstocks in breadfruit dwarfing [2]. During the development of the dwarf phenotype, breadfruit scions displayed significant increases in total flavonoid content, together with an increase in the expression of the flavonoid biosynthetic gene, *AaCHS* (Figure 1). These results are consistent with findings in mandarin scions growing on dwarfing rootstock ‘Flying Dragon’ [17] and pear scions growing on dwarfing interstocks [16] where flavonoid metabolites were significantly upregulated when compared with those on vigorous rootstocks/interstocks. Our results are also in agreement with previous findings on dwarfing apple rootstocks where flavonoid biosynthetic gene upregulation, accompanied by higher flavonoid content, was shown in scions growing on M.9 and M.27 dwarfing rootstocks when compared with those on vigorous rootstocks [15]. Metabolome analysis also revealed that increased flavonoid content in pear scions was involved in the reduction of scion growth caused by dwarfing interstocks [16]. Furthermore, an MYB gene involved in the early stages of flavonol biosynthesis was upregulated in sweet cherry scions during the differential cessation of terminal meristem growth caused by dwarfing rootstocks [18]. The current results are also consistent with a previous RNA-sequencing analysis showing the upregulation of flavonoid biosynthetic genes in breadfruit scions growing on another dwarfing-related rootstock, marang [10]. Flavonoids have long been proposed as inhibitors of polar auxin transport [14,19]. Over-accumulation of flavonoids due to mutations in genes encoding enzymes or transcription factors involved in flavonoid biosynthesis causes dwarf phenotypes [14,19,44,45,46]. This evidence suggests that increased flavonoid content may contribute to the dwarf stature observed in breadfruit scions growing on lakoocha rootstocks. Correlation coefficient analysis further suggested a strong, positive correlation between the transcript levels of *AaCHS* and the total flavonoid contents in the tested period, *r* = +0.8623, *p* < 0.001 (Appendix A). The results are in agreement with a previously reported relationship between *CHS* expression and total flavonoid content in many species, such as citrus [23], apple [47], walnut (*Juglans regia*) [25,26], and licorice (*Glycyrrhiza uralensis*) [48]. CHS is generally encoded by a multigene family [24,39]; the *AaCHS* gene in breadfruit in our study, when compared with the large size of the *CHS* family in other species, such as mulberry (*Morus atropurpurea*) with five *MaCHS* genes [36], grapevine (Vitis Vinifera) with three *VvCHS* genes [37], apple (Malus × domestica) with three *MdCHS* genes [38], and citrus (*Clementina Oroval* × *Citrus*. *sinensis* ‘Moro’) with at least three *CitCHS* genes [23], suggests that other breadfruit *CHS* genes remain to be cloned. However, the strong correlation between the expression of *AaCHS* and total flavonoid levels suggests that the transcriptional regulation of the *AaCHS* gene is involved in the control of flavonoid biosynthesis in breadfruit scions in response to lakoocha rootstocks. On the other hand, the fact that transcript levels of the dihydroflavonol 4-reductase gene, *AaDFR*, did not change across scions on different rootstocks suggests the transcription of *AaDFR* has less of an impact on the increased total flavonoid content. DFR is a key committed enzyme at the branching point of the flavonoid pathway, leading to the production of anthocyanin [20]. The expression of *DFR* genes was reported to highly correlate with anthocyanin accumulation [30,32,33,34]. It remains to be investigated whether upregulated *CHS* expression, coupled with unchanged DFR expression, would lead to a higher proportion of early flavonoid products in scions on lakoocha rootstocks. Taken together, the current results suggest that increased total flavonoid content, together with upregulated *AaCHS*, may play a role in the growth inhibition of breadfruit scions growing on lakoocha rootstocks.

Flavonoids represent an important mechanism in mediating plant growth and stress-induced morphogenetic responses [21,22]. The biosynthesis of flavonoids is controlled by a range of endogenous and environmental signals [21]. Notably, flavonoid biosynthetic genes, including *CHS* genes, have been shown to be activated by sucrose [49,50]. Significantly higher levels of sucrose were indeed found in breadfruit scions growing on lakoocha rootstocks in comparison to those grown on self-grafts or non-grafts [2]. Upregulation of flavonoid biosynthetic genes, accompanied by higher sucrose levels, was also found in breadfruit scions growing on another dwarfing-related rootstock, marang [10]. This points to a potential role of disruption in sucrose utilization, and its impact on flavonoid biosynthesis and growth inhibition. On the other hand, flavonoid biosynthesis is under the cooperative regulation of lignin biosynthesis [51]. Increased flavonoid biosynthesis is associated with the redirection of metabolic flux away from lignin biosynthesis in scions on apple dwarfing rootstocks [15]. A previous study also showed dwarfing of breadfruits through marang rootstocks involved downregulation of lignin biosynthetic genes [10]. Lignin biosynthesis responses to lakoocha rootstocks remain unknown. With both derived from the general phenylpropanoid pathway, the biosynthesis of flavonoid and lignin is regulated by the MBW complex, which is composed of MYB transcription factors, basic helix–loop–helix (bHLH) factors, and WD40 proteins [21]. MYB transcription factors are under the global regulation of diverse signal pathways, including ABA, jasmonate (JA), auxin, and GA [21]. These signaling pathways have long been proposed to be involved in rootstock-related dwarfism [3,4]. For example, apple dwarfing rootstock M.9 transports less IAA and more ABA than the vigorous rootstock [5]. Elevated ABA levels in scions, leading to low hydraulic conductivity in the vascular system, have been proposed as part of the mechanism for rootstock-induced dwarfing in apple trees [52]. A total of 38 MYB genes were differentially expressed in breadfruit scions during the development of a dwarf phenotype induced by marang rootstocks [10]. Breadfruit scions also displayed downregulation of genes for auxin signaling, including auxin efflux carrier PIN3 and auxin response factor ARF, but upregulation of genes participating in production of bioactive jasmonyl-isoleucine (JA-Ile) when growing on dwarfing rootstock, marang [53]. Notably, DELLA proteins (GA signaling repressors) were over-accumulated in these breadfruit scions [53]. DELLA proteins have been found to interact with MYB transcription factors to promote flavonol biosynthesis [21]. The role of flavonoids and their interactions with other signal regulators in growth inhibition induced by lakoocha rootstocks deserves further investigation.

In conclusion, breadfruit scion stems growing on lakoocha rootstocks displayed significant increases in total flavonoid content during dwarf phenotype development. The over-accumulation of flavonoids was correlated with the upregulation of *AaCHS*. This work provides insight into the functional significance of flavonoid biosynthesis in the rootstock-induced development of dwarf phenotypes in breadfruit plants.

## 4. Materials and Methods

### 4.1. Plant Materials and Treatments

Breadfruit (*Artocarpus altilis* cv. Noli) rooted cuttings and lakoocha (*Artocarpus lakoocha*) seedlings were obtained from a nursery at Cairns, northern Queensland. All plants were grown in a glasshouse at 25 to 28 °C and were supplied with natural daylight and daily water. Plants were grown in pots with vermiculite, soil, and fertilizer as previously reported [54]. Scions selected from breadfruit plants of 30 to 50 cm tall were grafted onto lakoocha rootstocks using approach grafting [55]. For self-graft controls, breadfruit scions were grafted onto rootstocks of the same breadfruit cultivars. At six months after grafting, each established grafted plant was gradually transferred to a bigger pot, until an 85-litre pot was required, and continued to grow under the same conditions. For growth comparison, own-rooted breadfruit plants (non-grafts) were grown alongside under the same condition. Interspecific grafts were constantly monitored for graft compatibility. Only compatible grafts were used for further analysis. Compatible grafts were defined as: (1) scions displaying active growth with continuous emergence of green leaves from the apex, and progressive development of new internodes from apical buds or axillary buds; (2) no fissures identified through visual inspection (Appendix A).

### 4.2. Determination of Total Flavonoid Content

Stem tissues (100 mg) were triple-extracted in 80% ethanol at room temperature for 2 h. Combined supernatants were used for measurement of total flavonoid content based on an aluminum chloride colorimetric method [56], with slight modification. Briefly, 0.5 mL of supernatant was mixed with 0.1 mL of AlCl_3_ solution (10%, *w*/*v*), 0.1 mL of 1 M CH_3_COOK, and 2.3 mL of deionized water. After incubation at room temperature for 30 min, absorbance at λ = 426 was measured. The concentration of total flavonoids was expressed as mg of quercetin equivalent (QE) per gram dry weight (DW).

### 4.3. Cloning Chalcone Synthase and Dihydroflavonol 4-Reductase cDNA

Total RNA of breadfruit scion stems was extracted using an RNeasy kit with on-column DNase digestion (Qiagen, Clayton VIC, Australia). In order to isolate the full-length genes encoding chalcone synthase and dihydroflavonol 4-reductase, the extracted RNA was subjected to 5′ and 3′ SMART RACE RT-PCR followed by full-length amplification (Takara Clontech, CA, USA). PCR products cloned into a pGEMT vector (Promega, Sydney, Australia) were sequenced. Sequence information of the partial *CHS* and *DFR* [2], together with the 5′ or 3′ RACE PCR products, were used to design primers for full-length amplification. Five full-length clones of each gene were sequenced in double strands. The resulting sequences were analyzed with Sequencher (version 4.1, Gene Codes Corporation, MI, USA). Sequences of the breadfruit chalcone synthase gene *AaCHS* and the dihydroflavonol 4-reductase gene *AaDFR* reported here are available in GenBank (http://www.ncbi.nlm.nih.gov/) under accession numbers OP921038 and OP921039.

### 4.4. Quantitative Real-Time PCR

After separation from plants, stem tissues (second internode regions) were immediately immersed in RNA*later* (Life Technologies, Mulgrave VIC, Australia) and stored at −80 °C. Total RNA was extracted and reverse-transcribed using Superscript reverse transcriptase and oligo(dT) (Life Technologies, Australia). Real-time PCR was carried out on a Corbett Research Rotor-Gene 6000 cycler using the QuantiFast SYBR Green PCR Kit (Qiagen, Australia) as described [54]. Thermocycling was initiated with a 5 min incubation at 95 °C followed by 40 cycles (95 °C for 10 s; 60 °C for 30 s). Primer efficiency (see Appendix A for primer sequences) was examined using standard curves from cDNA samples with serial dilutions. Each reaction was performed in duplicate (technical repeat) with non-reverse-transcribed cDNA as a negative control (non-template control). Housekeeping genes, actin (*AaActin*) and elongation factor 1-α (*AaEFα-1*), were evaluated for their stability in scion stems as previously described [10]. The actin gene was chosen to normalize the expression of transcript abundance. The expression of each gene was an average of five biological replicates.

### 4.5. Statistical Analyses

Significant differences were determined with analysis of variance (ANOVA) followed by multiple comparisons using Tukey’s HSD at *p* < 0.05. All analyses were performed in SPSS Statistics version 27.

## Figures and Tables

**Figure 1 plants-12-03285-f001:**
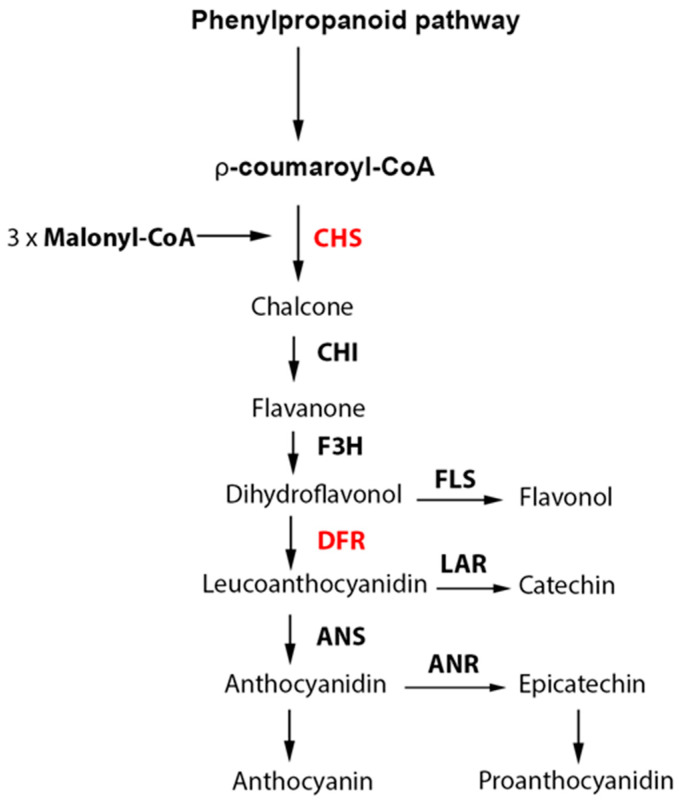
Schematic showing the flavonoid pathway. CHS, chalcone synthase; CHI, chalcone isomerase; F3H, flavanone 3-hydroxylase; FLS, flavonol synthase; DFR, dihydroflavonol 4-reductase; LAR, leucoanthocyanidin reductase; ANS, anthocyanidin synthase; ANR, anthocyanidin reductase.

**Figure 2 plants-12-03285-f002:**
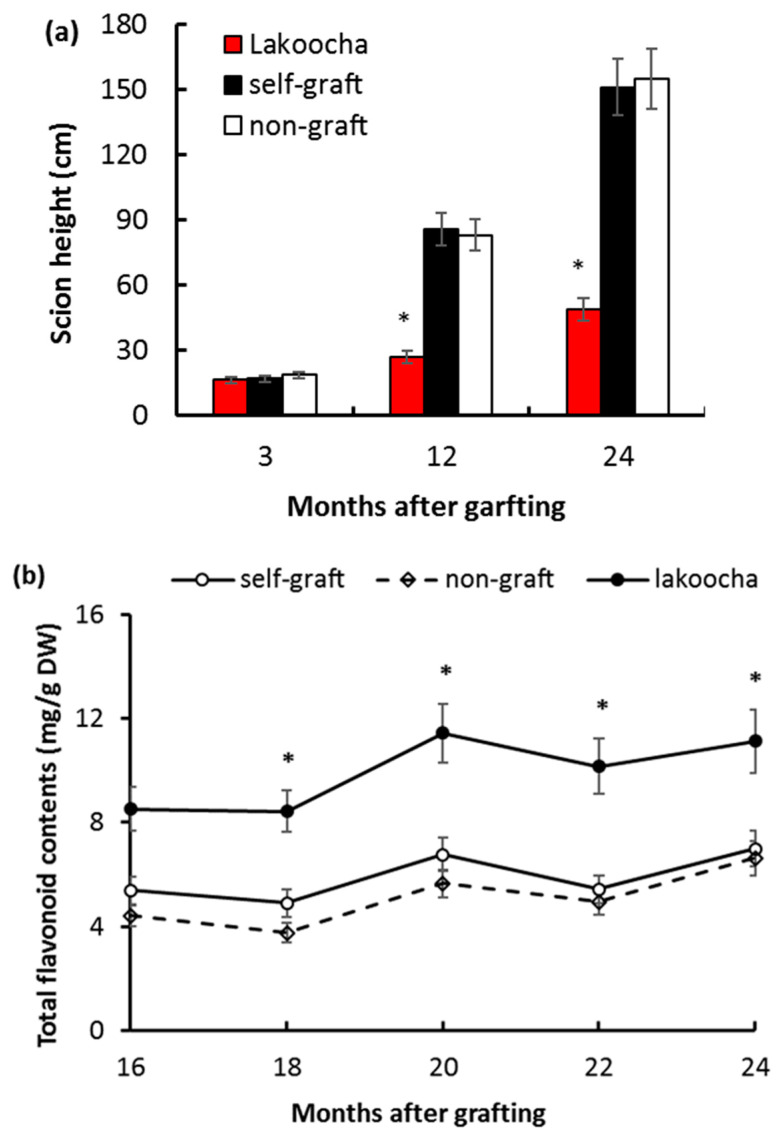
Rootstock effects on the phenotype of breadfruit scions. (**a**) Height of scion main stems on different rootstocks in the period up to 24 months after grafting. (**b**) Temporal profiles of total flavonoid content in scion stems for different rootstocks. All values represent mean ± SE from five biological replicates. * Significant difference (*p* < 0.05).

**Figure 3 plants-12-03285-f003:**
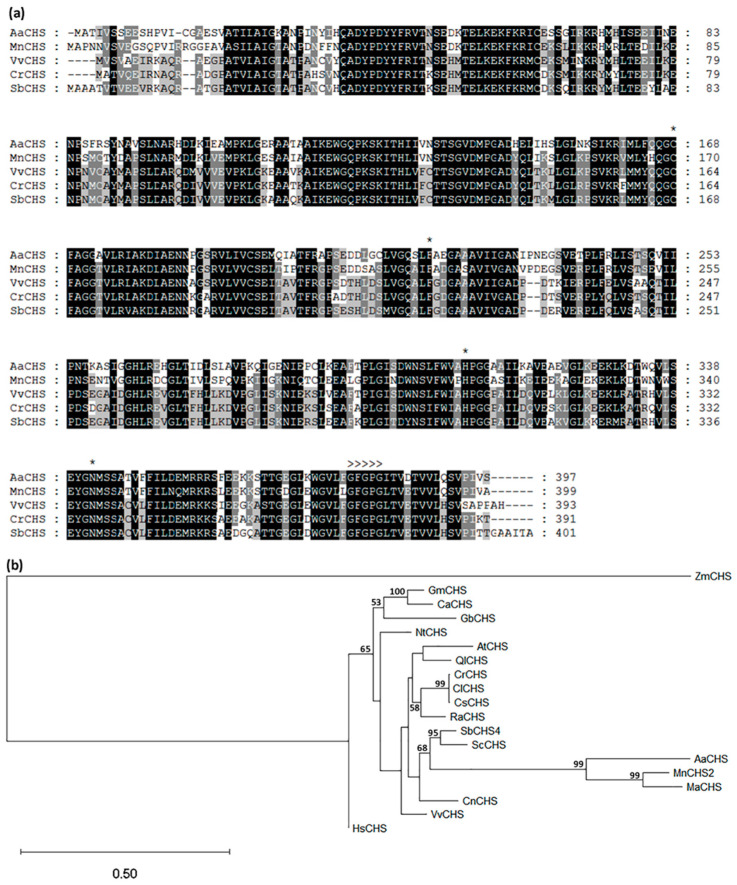
Amino acid sequence alignments and phylogenetic analysis of chalcone synthases from *Artocarpus altilis* and other plant species. (**a**) Amino acid sequence alignment of AaCHS from breadfruit (*Artocarpus altilis*) with MnCHS (*Morus notabilis*), VvCHS (*Vitis vinifera*), CrCHS (*Citrus reticulata*), and SbCHS (*Sorghum bicolor*). Sequence alignments were performed using the ClustalO algorithm. Shaded levels represent conservation degree. Annotations include: a catalytic site related to Cys164, Phe215, His303, and Asn336 (*), and the characteristic CHS signature, GFGPG (>>>>>). (**b**) phylogenetic analysis of AaCHS in relation to other plant chalcone synthases. A phylogenetic tree was constructed based on the maximum likelihood method in the PHYLIP package, and unrooted in MEGA software (version 11). Bootstrap values of more than 50% of 1000 replicates are shown next to nodes. All sequences were obtained from NCBI (https://www.ncbi.nlm.nih.gov, accessed on 12 November 2022). *Morus notabilis* MnCHS2 (XP_024027307); *Morus alba* MaCHS (AOV62760); *Citrus sinensis* CsCHS (ACB47461); *Citrus limon* MlCHS (AKR05585); *Citrus reticulata* CrCHS (AKR05589); *Sorghum bicolor* SbCHS4 (XP_002450870); *Vitis vinifera* VvCHS (RVW45074); *Zea mays* ZmCHS (NP_001149508); *Nicotiana tabacum* NtCHS (NP_001312634); *Glycine max* GmCHS (NP_001304585); *Ginkgo biloba* GbCHS (AAT68477); *Arabidopsis thaliana* AtCHS (AAB35812); *Secale cereal* ScCHS (CAA63306); *Cicer arietinum* CaCHS (CAA10131); *Hypericum sampsonii* HsHS (AFU52909); *Quercus lobate* QlCHS (XP_030953406); *Rhodamnia argentea* RaCHS (XP_030512195); *Cocos nucifera* CnCHS (KAG1370226).

**Figure 4 plants-12-03285-f004:**
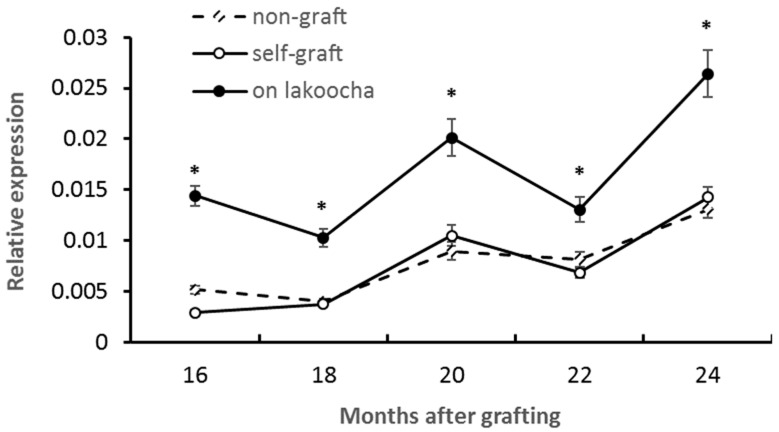
Temporal profiles of *AaCHS* expression in breadfruit scion stems in response to different rootstocks. Expression level of the transcript was normalized to the expression of the actin gene. All values represent mean ± SE from five separate RNA extractions (* *p* < 0.05).

**Figure 5 plants-12-03285-f005:**
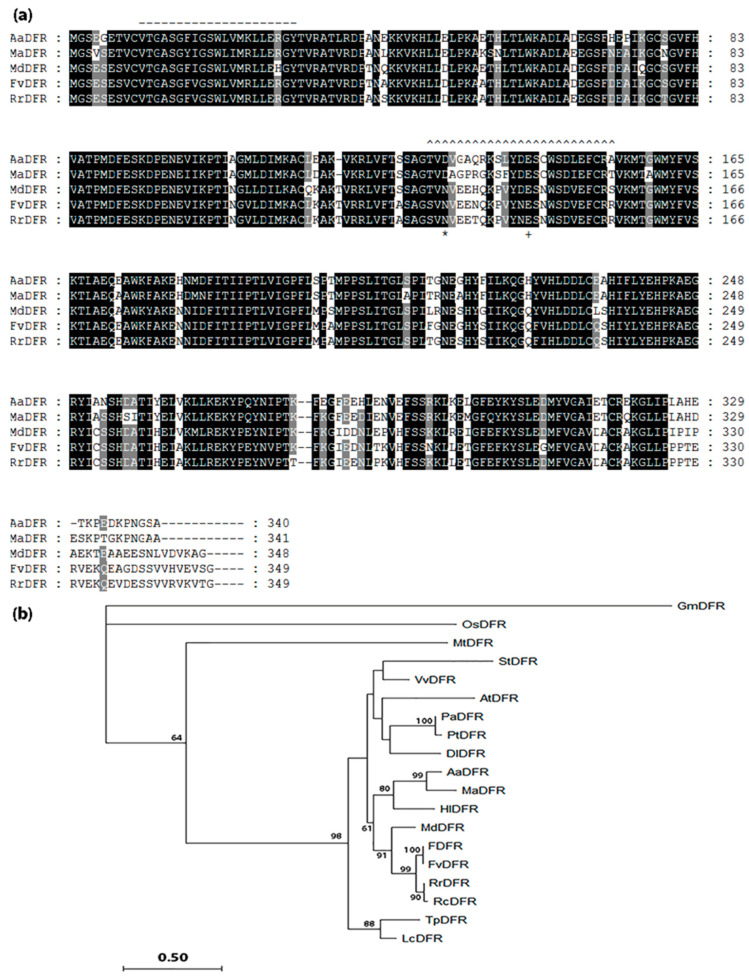
Amino acid sequence alignments and phylogenetic analysis of dihydroflavonol 4-reductases from *Artocarpus altilis* and other plant species. (**a**) Alignment of MaDFR (*Morus alba*), MdDFR (*Malus domestica*), FvDFR (*Fragaria vesca*), and RrDFR (*Rosa rugosa*), and deduced amino acid sequences of AaDFR from breadfruit scion stems. Sequence alignments were performed using the ClustalO algorithm. Shaded levels represent the conservation degree. Annotations include a putative NADPH-binding motif (------), presumed substrate-binding region (^^^^^), and substrate specificity associated with aa134 (*) and aa145 (+) [43]. (**b**) Phylogenetic analysis of AaDFR in relation to other plant dihydroflavonol 4-reductases. A phylogenetic tree was constructed based on the maximum likelihood method in the PHYLIP package, and unrooted in MEGA11 software. Bootstrap values of more than 50% of 1000 replicates are shown next to nodes. All sequences were obtained from NCBI (http://www.ncbi.nlm.nih.gov, accessed 25 October 2022). *Arabidopsis thaliana* AtDFR (CAP08826); *Dimocarpus longan* DlDFR (QRV61370); *Glycine max* GmDFR (NP_001236658); *Fragaria vesca* FvDFR (AHL46448); *Humulus lupulus* HlDFR (BAF45153); *Lotus corniculatus* LcDFR (AAV71171); *Malus domestica* MdDFR (AAO39816); *Medicago truncatula* MtDFR (AES79932); *Morus alba* MaDFR (AHB19302); *Oryza sativa* OsDFR (BAC78578); *Populus alba* PaDFR (TKS10150); *Populus tremuloides* PtDFR (AAN63056); *Rosa chinensis* RcDFR (XP_024167119); *Rosa rugose* RrDFR (AIU34714); *Solanum tuberosum* StDFR (AAZ57436); *Trifolium pratense* TpDFR (PNX92286); *Vitis Vinifera* VvDFR (CAA53578).

**Figure 6 plants-12-03285-f006:**
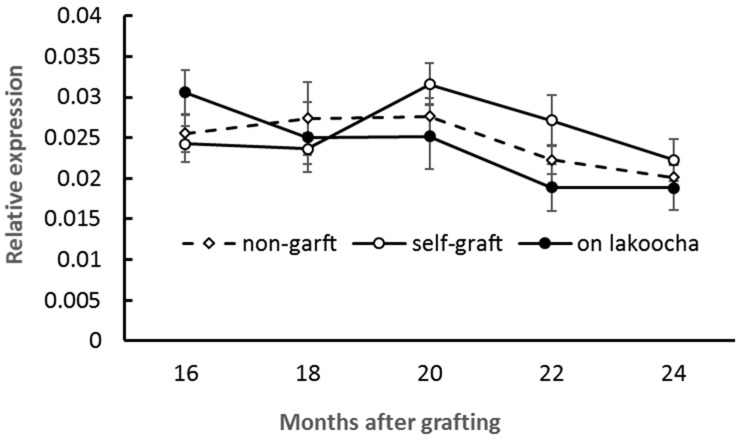
Temporal profiles of *AaDFR* expression in breadfruit scion stems in response to different rootstocks. Expression level of the transcript was normalized to actin gene expression. All values represent mean ± SE from five separate RNA extractions.

## Data Availability

Sequences of the breadfruit chalcone synthase gene *AaCHS* and dihydroflavonol 4-reductase gene *AaDFR* reported in this article are available in GenBank (http://www.ncbi.nlm.nih.gov/) under accession numbers OP921038 and OP921039.

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
