# Peer review of "Total Flavonoid Contents and the Expression of Flavonoid Biosynthetic Genes in Breadfruit (Artocarpus altilis) Scions Growing on Lakoocha (Artocarpus lakoocha) Rootstocks"

_plants, 2023, doi:10.3390/plants12183285_

Round 1

Reviewer 1 Report

The manuscript entitled "Expression of flavonoid biosynthetic genes and total flavonoid 2 contents in breadfruit (Artocarpus altilis) scion stems growing 3 on lakoocha (Artocarpus ) rootstocks" focused on the the significance of 21 flavonoid biosynthesis in rootstock-induced breadfruit dwarfing. Overall, manuscript is well-written and could be accepted for publication. Authors can do following two changes during proof-reading. 

L382-383: please re-write

Table S1: Better to mention as "reverse primers"

Few minor mistakes

Author Response

Review 1

Comments and Suggestions for Authors

The manuscript entitled "Expression of flavonoid biosynthetic genes and total flavonoid 2 contents in breadfruit (Artocarpus altilis) scion stems growing 3 on lakoocha (Artocarpus ) rootstocks" focused on the the significance of 21 flavonoid biosynthesis in rootstock-induced breadfruit dwarfing. Overall, manuscript is well-written and could be accepted for publication. Authors can do following two changes during proof-reading.

L382-383: please re-write

Reply:  We are glad that the reviewer recognises the merit of our effort.   The related sentence has now been re-written (see highlighted in yellow).

Table S1: Better to mention as "reverse primers"

Reply: This has been corrected.

 Comments on the Quality of English Language

Few minor mistakes

Reply:  The paper has been fully checked to correct all minor mistakes.

Reviewer 2 Report

The manuscript proposes a nice piece of work and is well-written, however, the authors have not described their results fully in the discussion section, which needs to be elaborated well, quoting the latest and most relevant literature on this subject.

The methodology is written concisely and results are correctly placed.

I suggest incorporating some of more latest references in the introduction and discussion sections for further understanding and covering the topic.

The manuscript proposes a nice piece of work and is well-written, however, the authors have not described their results fully in the discussion section, which needs to be elaborated well, quoting the latest and most relevant literature on this subject.

The methodology is written concisely and results are correctly placed.

I suggest incorporating some of more latest references in the introduction and discussion sections for further understanding and covering the topic.

Author Response

Review 2

The manuscript proposes a nice piece of work and is well-written, however, the authors have not described their results fully in the discussion section, which needs to be elaborated well, quoting the latest and most relevant literature on this subject.

The methodology is written concisely and results are correctly placed.

I suggest incorporating some of more latest references in the introduction and discussion sections for further understanding and covering the topic.

Reply

We are glad that the reviewer recognises the merit of our effort, and provides constructive advice to improve our manuscript.  We have now added more information into the Introduction and more in-depth discussion with some latest relevant references.  We have also checked all the references to make sure there are relevant.  We have modified the Method section. These added sections are highlighted in yellow.  

Minor editing of English language required

Reply: The paper has been fully checked to correct all minor mistakes.

Reviewer 3 Report

1. In Abstract,15-30 m tall.

2. 65% reduction in breadfruit tree size: size or height?

3. in results: Breadfruit plants on lakoocha rootstocks displayed shorter stature with height reduced by 68.6% at 12 months and by 67.7% at 24 months after grafting compared to those on self-graft at the same time. size should be height.

4. What is the quantitative relationship between flavonoid contents and height?

5. In sentence: Transcript levels of AaCHS were analysed using quantitative real-time RT-PCR in scion stems in relation to different rootstocks. different rootstocks should be different times rootstocks?

6. 2.3. Isolation and expression analysis of dihydroflavonol 4-reductase gene in breadfruit scions.

this part should be simplified.

7. In discussion: Breadfruit plants with lakoocha rootstocks were about one-third of the standard height at the end of 24 months after grafting (Figure 1), Where is the picture?

8. A photograph of the graft should be added to the paper.

9. The title of the paper is unclear and should be revised

Minor editing of English language required。

Author Response

Review 3

Comments and Suggestions for Authors

  1. In Abstract,15-30 m tall.

Reply: This has been corrected.

  1. 65% reduction in breadfruit tree size: size or height?

Reply:  It is 65% reduction in height. In the current study, only tree height data was presented.  However, lakoocha rootstocks also led to reduction in stem thickness with shorter internodes, fewer branches, fewer and smaller leaves, resulted in a smaller tree size [1]. 

  1. in results: Breadfruit plants on lakoocha rootstocks displayed shorter stature with height reduced by 68.6% at 12 months and by 67.7% at 24 months after grafting compared to those on self-graft at the same time. size should be height.

Reply:  These are tree height data.  However, previous work has suggested lakoocha rootstock has the potential to control breadfruit tree vigour, leading to a smaller size tree (see answer above). Accordingly, some sentences have been re-worded to accurately address the topics.

  1. What is the quantitative relationship between flavonoid contents and height?

Reply:  According to Figure 2, lakoocha graft had higher levels of flavonoid content compared to those of self-graft and non-graft, however, there was no significant change of flavonoid levels across the period from 16 to 24 months for any rootstock type.  For instance, the flavonoid contents at 24 month was not significantly higher than those at 16 months for any rootstock type, even though all plants grew taller as time progressed.  More investigation is needed to draw a conclusion on the correlation between flavonoid contents and height.

  1. In sentence: Transcript levels of AaCHS were analysed using quantitative real-time RT-PCR in scion stems in relation to different rootstocks. different rootstocks should be different times rootstocks?

Reply: this sentence has been re-written.

  1. 2.3. Isolation and expression analysis of dihydroflavonol 4-reductase gene in breadfruit scions.

this part should be simplified.

Reply: We have simplified this section.

  1. In discussion: Breadfruit plants with lakoocha rootstocks were about one-third of the standard height at the end of 24 months after grafting (Figure 1), Where is the picture?

Reply:  The conclusion was calculated based on the actual height measurement (see Figure 2). We have now also added a picture (see Figure S1) in the revised manuscript as suggested by the reviewer.

  1. A photograph of the graft should be added to the paper.

Reply: 

Photographs of the graft, showing different part of the grafted plants have now been added (see Figure S3) as recommended by the reviewer.  However, photographs of more detailed morphological characterisation of the graft, and its time-course development have recently published [1].  

  1. The title of the paper is unclear and should be revised

Reply: We have revised the title.  The modified title is ‘Total flavonoid contents and the expression of flavonoid biosynthetic genes in breadfruit (Artocarpus altilis) scions growing on lakoocha (Artocarpus lakoocha) rootstocks’,

Minor editing of English language required

Reply:  The revised manuscript has been fully checked to correct all minor mistakes.

  1. Zhou, Y.; Underhill, S.J.R. Characterisation of breadfruit (Artocarpus altilis) plants growing on Lakoocha (A. lakoocha) rootstocks. Horticulturae 2022, 8,916

Reviewer 4 Report

This article describes in detail the expression of flavonoids contained in breadfruit, article described in detail. The strong point is represented by the clear conclusions where it provides insight into the functional significance of flavonoid biosynthesis in the development of dwarf phenotype in breadfruit plants through lakoocha rootstocks.

Author Response

Review 4

Comments and Suggestions for Authors

Comments and Suggestions for Authors

This article describes in detail the expression of flavonoids contained in breadfruit, article described in detail. The strong point is represented by the clear conclusions where it provides insight into the functional significance of flavonoid biosynthesis in the development of dwarf phenotype in breadfruit plants through lakoocha rootstocks.

 Reply:  We are glad the reviewer appreciates the premise and results of the manuscript.